# Mineral Constituents Profiling of Ready-To-Drink Nutritional Supplements by Inductively Coupled Plasma Optical Emission Spectrometry

**DOI:** 10.3390/molecules25040851

**Published:** 2020-02-14

**Authors:** Anna Leśniewicz, Daniela Kurowska, Paweł Pohl

**Affiliations:** Analytical Chemistry and Chemical Metallurgy Division, Faculty of Chemistry, Wrocław University of Science and Technology, Wybrzeże St. Wyspiańskiego 27, 50-370 Wrocław, Poland; aniela.kurowska@gmail.com

**Keywords:** medical nutrition, nutritional drinks, lyophilization, freeze-drying, mineral constituents, ICP-OES, interelement correlations

## Abstract

Nutritional drinks (NDs) are medicinal food products intended for people with different health issues constricting nutrients provision. Eight varieties of milkshake style NDs were analyzed in this work. Prior to element analysis, they were freeze-dried, and concentrations of twenty macro- and microelements in analyzed samples were simultaneously measured by ICP-OES after their mineralization in a closed-vessel microwave-assisted digestion system. Results of this analysis indicated that these NDs must be considered as nutrient-dense foods, taking into account mineral constituents. Consumption of two bottles of such NDs per day provides very a high amount or even an excess of human daily requirements set as Recommended Dietary Allowances (RDAs). Generally, concentrations of determined elements in examined NDs were consistent with data given on the labels—most of differences did not exceed 30% (median: −5.91%, standard deviation: 14%). Discovered very strong and moderate positive correlations between concentrations of major and essential elements (Ca, Mg, P, Cu, Fe, Mn, Zn) were likely due to their incorporation into formulations of analyzed NDs. However, relationships between contents of trace elements were the result of concomitance of these elements in substrates used for examined products production or contamination of substrates.

## 1. Introduction

According to dietitian nutritionists, good nutrition, based on an adequate, well-balanced and diversified diet, combined with regular physical activity is foundation of good health. On the other hand, an unhealthy diet is the major risk factor for a number of chronic, non-communicable diseases [1]. Unfortunately, there are several circumstances, such as disease, distress, stress and others, in which the patient’s body cannot obtain sufficient nutrients through the diet. In those situations, dietary supplementation, particularly formulated for certain patient conditions, is highly required to his proper functioning and well-being. Among different products aimed at administrating liquid oral nutritional supplements, there are hypoallergenic products for babies and children with milk allergy or allergy to multiple dietary proteins, as well as nutrition products adapted to diet management of patients who are unable to normally or sufficiently eat.

The interest in regular consumption of nutritional drinks (NDs) seems to be increasing, and potential consumers should be able to choose the highest-quality products that have special properties and are nutritionally safe. Unfortunately, none of European countries has restricted the use of health claims to foods of high nutritional quality despite the regulatory background provided by the European Union in 2006 [2]. For that reason, it is mandatory to reformulate food products available in the food supply because it can improve unhealthy diets and reduce the public health burden related to non-communicable diseases. This is certainly not possible without a methodical monitoring of the food supply in reference to a variety of aspects. Results of such studies can contribute to development of appropriate legislation criteria for nutritional quality as well as build a comprehensive database, enabling to differentiate between niche and market-leading products.

To the best of our knowledge, there are only few reports on determination of elements in foods for medical purposes in Poland [3,4,5]. These articles report the content of selected metals in prescription food products for special medical purposes and modified milk products for newborns and infants. There are also single publications on determination of certain minerals in enteral nutrition formulas from Brazil [6] and Spain [7]. In published works, the content of such main elements as Ca, Mg, P, Na, K [6,7], or essential trace elements, i.e., Cu, Mn, Fe and Zn [5,6,7], and potentially toxic metals, i.e., Cd, Cr, Ni and Pb [3,4,7], were quantified. Most often, total concentrations of elements were assessed [3,4,5,6,7], but the dialyzable fraction of Ca, Mg, Fe, Cu and Mn was additionally apportioned [7]. Prior to element analysis, medicinal food samples were mineralized in concentrated nitric acid [3,4,5,6] in addition to hydrogen peroxide [6]. Typically, closed-vessel, microwave-assisted wet digestion was used for decomposition of samples as received, or tedious and long-lasting dry ashing was applied, providing additional pre-concentration of trace elements [7]. As far as we know, no studies on simultaneous determination of twenty main and trace elements in medical nutrition products available in pharmacies for multitude of consumers have been undertaken and reported.

The main goal of this work was the mineral composition profiling of selected commercial milk shake-style NDs and comparison of results with producer’s declarations as the important part of quality control and monitoring of these food supplements. Popular NDs brands, widely distributed on the international market, were selected for this investigation. Their mineral composition was based on quantification of given elements (B, Ca, Cd, Co, Cr, Cu, Fe, Mg, Mn, Mo, P, Ti, V and Zn) and was extended by the information on elements (Ag, Al, B, Ba, Ni and Sr) for which no data were claimed on labels. The additional purpose of this work was evaluation of the nutritional value of these food supplements, considering recognized requirements and daily permissible doses.

## 2. Results and Discussion

### 2.1. Trueness of the Analytical Method

Trueness and precision of the microwave-assisted wet digestion procedure and measurements by the inductively coupled plasma optical emission spectrometry (ICP-OES) method were evaluated by analysis of an adequate certified reference material (CRM), i.e., NIST non-fat milk powder SRM 1549. Results of this analysis are shown in Table 1. In the last column of this table, outcomes of the two-tailed *t*-test (α = 0.05, df = 3 + 3 – 2 = 4, t*_critical_* = 2.776) with indication of statistically significant differences or not (S or N) are shown.

Comparison of experimental results with values given in the CRM attest proved good trueness of the applied sample preparation procedure, i.e., closed-vessel microwave-assisted wet digestion combined with ICP-OES. For trace elements, i.e., Al, Cu, Fe, Mn, and Mo, differences between certified and measured concentrations were lower than 10%. For those elements, their recoveries amounted to 106%, 107%, 101%, 92.3%, and 97.1%, respectively. In case of other elements (major Ca and Mg, minor Zn), agreement between measured and certified concentrations was also good because their recoveries were 95.4, 90.8, and 98.3%, respectively. In all cases, the *t*-test indicated that differences between determined and certified concentrations of studied elements were statistically insignificant at the 95% confidence level. Acceptably good precision of experimental results was shown by relative standard deviation (RSD) values, which were in the range from 1.8% to 7.3% for Ca, Mg, and Zn, and from 4.2% to 11.7% for Al, Cu, Fe, Mn and Mo. Both measures achieved indicated that the proposed methodology was suitable for the purpose of this work in relation to major, minor and trace elements.

### 2.2. Concentrations of Macro-, Micro- and Trace Elements in the Nutritional Drinks

Total concentrations of studied elements (Ag, Al, B, Ba, Ca, Cd, Co, Cr, Cu, Fe, Mg, Mn, Mo, Ni, P, Pb, Sr, Ti, V and Zn) in examined NDs, i.e., ND1-ND8, are shown in Table 2. The content of elements is given as mean values along with their standard deviations (SDs) and expressed in mg of the mineral constituent in 100 mL of the original ND sample. The concentration of every element was calculated based on triplicate measurements performed for three parallel samples of each ND (*n* = 9).

Before determination of elements by using the methodology described above, examined ND samples were freeze-dried. It was established that the mean mass loss ranged from 52.7% to 76.6%. The highest water content was noticed in ND8, while the lowest mass decrement during the vacuum freeze-drying process was observed for ND5. The mean water content in samples ND1-ND6, including those enriched in proteins (ND5) and fibers (ND6), was 54.5%. There was almost no difference in the water content in case of these NDs, because coefficient of variation (CV) was just 3.1%. Only ND7 and ND8 samples contained over 70% of water (73.1% and 76.6%, respectively). Based on these results, it was recognized that freeze-drying of analyzed milk shake-style NDs (ND1-ND8) at the stage of sample preparation seemed to be quite convenient; samples were homogenized while trace elements were pre-concentrated prior to ICP-OES measurements with an enrichment factor within the range of 2.2–4.3.

Comparison of results of element analysis of all ND samples clearly indicated that the smallest number of elements was determined in ND7 and ND8. The content of B, Cd, Cr, Ni, Pb and V in these nutritional supplements was lower than respective limits of quantification (LOQs), i.e., 0.067, 0.0003, 0.0005, 0.003, 0.008 and 0.0008 mg/100 mL. In the nutritional drink of neutral flavor (ND1), maximal concentrations of micro- and trace elements, i.e., B, Fe, Mo, Ni and Ti, were noted. In general, concentrations of macroelements measured in all examined NDs were in agreement with those reported previously for milk-derived products. For example, Ca and Mg content ranges were similar to values reported for milk and yoghurt samples from Croatia [8], milk and fermented dairy products from Turkey [9], or milk and milk products from China [10]. Moreover, findings for Fe and Zn were comparable to concertation ranges established for milk and dairy products given in literature (see for example [10]). Jurowski and his colleagues [5] obtained similar results for Zn in case of Polish medicinal food products intended for infants. Concentrations of the rest of quantified mineral constituents in studied NDs were at most close to (Mn, B and Cu) or lower than 0.5 mg per 100 mL. Their content corresponded to values previously reported for milk products [10] and hypoallergenic infant formulas form the European market [11]. However, concentrations of Cu and Mn in Polish milk-based medicinal foods for newborns and infants [5] were lower than our findings, but still within the same order of magnitude. Interestingly, the content of all essential elements determined in commercially available Spanish infant formulas, from starter to lactose free and soya formulas [12], were similar to corresponding results obtained in our work.

Among all examined elements, the highest concentrations were observed for Ca followed by P. Determined concentrations of these two macroelements varied from 56.1 mg/100 mL (ND8) to 269 mg/100 mL (ND5) and from 67.2 mg/100 mL (ND8) to 332 mg/100 mL (ND5), respectively. Relatively high contents were also noticed for Mg, Fe, and Zn. The Mg content ranged from 26.3 mg/100 mL (ND8) to 269 mg/100 mL (ND5). The content of Fe and Zn did not vary much between examined NDs. The Fe content changed from 3.87 mg/100 mL (ND1) to 1.68 mg/100 mL (ND8). Concentrations of Zn were in the range from 4.85 mg/100 mL (ND7) to 1.21 mg/100 mL (ND8).

Considering the content of toxic metals, i.e., Cd, Cr, Ni, Pb, their presence in food products, especially in medicinal nutrition products, is detrimental and disqualifies their use for safe human administration. In all analyzed NDs, Cd, Cr, Pb and V was not quantified; their concentrations were lower than respective LOQs, i.e., 0.0003, 0.0005, 0.008 and 0.0008 mg/100 mL. These results were consistent with findings of Jurowski et al. [3,4] for Polish medicinal food products for infants, who quantified Cd, Cr and Pb in medical food samples and modified milk samples within the following ranges: 0.0008–0.0012, 0.0006–0.0235 and 0.0011–0.0024 mg/100 g. They also fulfilled the respective EU commission regulation [13] about the need for the Cd content in liquid formulas manufactured from cows’ milk proteins or protein hydrolysates. Ni concentrations were established to change from 0.063 (ND1) to 0.028 (ND3) mg/100 mL. This was acceptable because Ni up to 0.038 mg/100 g was previously determined in other Polish medicinal foods for children [4], and from 0.023 to 0.137 mg/100 g in flavored yogurt drinks [14].

Concentrations of remaining elements were in the two following ranges: 0.003–0.055 mg/100 mL (Ag, Ba, Co, Mo, Ti), or 0.042–0.800 mg/100 mL (Al, B, Sr). These concentration ranges were comparable to those previously reported for milk, milk powders and other dairy products [15,16,17].

Precision of measurements of analyzed NDs samples was good. For most of examined elements, RSDs calculated for individual samples were found to be within 0.5% to 10%. Only in case of elements determined in very low quantities, e.g., Ag, Al, Ba, Ni, Sr and Ti, respective RSD values were higher, i.e., up to 30% and more. Nevertheless, it was acceptable and predictable for such low quantities of elements present in such a complex sample matrix.

### 2.3. Inter-Element Relationships

Linear relationships between elements concentrations in all examined nutritional drinks were investigated by calculating Pearson correlation coefficients (r). Very strong positive relationships, indicated by r values higher than 0.7, were noticed for the following pairs of elements: B-Ag (r = 0.711), Mo-Ag (r = 0.780), Mo-B (r = 0.706), Ti-B (r = 0.786), Mg-Ca (r = 0.961), P-Ca (r = 0.922), Sr-Ca (r = 0.831), Fe-Cu (r = 0.890), P-Mg (r = 0.796), Zn-Mg (r = 0.725), Zn-Mn (r = 0.954), Ti-Mo (r = 0.744) and Sr-P (r = 0.885). Strong positive associations (0.7 < r < 0.5) between elements contents were discovered for Cu-Ag (r = 0.581), P-Ag (r = 0.572), Ti-Ag (r = 0.564), Zn-Ca (r = 0.631), Mn-Cu (r = 0.606), Mo-Cu (r = 0.592), P-Cu (r = 0.598), Zn-Cu (r = 0.571), Mn-Fe (r = 0.606), Mo-Fe (r = 0.535), Zn-Fe (r = 0.505), Mn-Mg (r = 0.521), Sr-Mg (r = 0.687) and Zn-P (r = 0.545). Very strong negative correlations were established between Mn and Al (r = –0.801), and Zn and Al (r = –0.711), while strong negative interrelationships were found for pairs of Ba-B (r = –0.604) and Sr-B (r = –0.506).

Moderate positive linear relationships (0.5 < r < 0.3) between concentrations of studied elements occurred for the following pairs: Ca-Ag, Fe-Ag, Mg-Ag, Sr-Ag, Zn-Ag, Ba-Al, Ti-Al, Fe-B, Mn-B, Zn-B, Sr-Ba, Cu-Ca, Mn-Ca, Ti-Cu, P-Mn, P-Mo and Ti-P. By contrast, a moderate negative interrelationship between concentrations of Fe and Al was only found. No relationships or week correlations were established for the rest of examined elements. It was presumed that most of noticed correlations emerged due to intentional incorporation of elements essential for proper functioning of the human body into formulations of analyzed NDs. However, relationships between trace elements contents as well as trace elements and macro- and microelements were likely the result of elements concomitance or contamination of substrates used for examined products production.

### 2.4. Nutritional Value Compliance Assessment

To assess compliance of experimental results of element analysis of NDs with the content of given elements (Ca, Cu, Cr, Fe, Mg, Mn, Mo, P and Zn) declared by the producer and reported on labels, both sets of results were compared as shown in Table 3. The producer of examined medicinal food products stated that they contain from 0.012 (ND8) to 0.016 (ND1–4; ND6) mg/100 mL of Cr.

Results of this work indicated that none of examined NDs contained more than 0.0005 mg/100 mL of Cr. In case of other elements, it appeared that results of their determinations in examined NDs were consistent with data given on labels for all products. With respect to all elements (except for Mo), more than half of declared concentrations of elements (38 out of 56, 68%) were higher than corresponding concentrations determined in the study. It was particularly the case of ND1–ND6. Contrary, in ND7 and ND8 declared concentrations of Ca, Cu, Fe, Mg, Mn, P and Zn were lower than determined values. Differences between declared and determined values in overwhelming cases did not exceed 30%, i.e., they were from −17.2% to +28.9% for Ca, from −39.4% to +1.9% for Cu, from −12.1% to +19.0% for Fe, from −11.5% to +29.5% for Mg, from −19.8% to +30.3% form Mn, from −10.3% to +19.4% for P and from −31.4% to +7.8% for Zn. Nevertheless, it pointed out that analyzed ND products did not meet declared values and hence, they were not consistent with given description. The best agreement between declared and measured elements concentrations was obtained for Fe and P. The cumulative error considering all samples for these two elements was 25.9% (Fe) and 31.4% (P). In case of Ca, Cu, Fe, Mg, Mn, and P determined in ND1–ND6, mentioned differences between their declared and determined concentration were less than 20%.

The highest discrepancy between declared and measured concentrations of elements in examined NDs was established for Mo. In this case, measured Mo concentrations were much higher than their declared values, and differences were within the range from 54.2% (ND6) up to 220% (ND8). This certainly could be related to a very low concentration of this element in analyzed products, and a higher susceptibility to concentration changes in this case. Differences noted between mineral constituent contents declared by medicinal food producer on product labels and contents experimentally ascertained during their analysis were already reported by other researchers [5,7]. This is quite dangerous, especially when related not only to the content of macroelements, but also micro- and trace elements, or products sold in pharmacies on prescriptions.

### 2.5. The Contribution of Nutritional Drink Administration to Recommended Daily Allowances of Elements

The information on concentrations of major, minor and trace elements in analyzed NDs was important in reference to reliable assessment of their quality, safety and nutritive value. The safety and nutritive value of examined NDs was estimated by comparison of the intake of given elements caused by administration of their two bottles with tolerable uptake upper levels (TUULs) and recommended dietary allowances (RDAs) settled for these elements [18,19,20,21]. Results of this comparison are given in Table 4. Assessed data undoubtedly indicated that consumption of two bottles of NDs could provide very high amounts or even excess of daily human needs. However, it should be clearly stated that examined NDs are very dense medicinal food products intended for people with malnutrition caused by different health issues. It should not be either forgotten that differences between total contents of elements and their bioavailable fractions could be substantial. Nevertheless, according to the Regulation No 1169/2011 of the European Parliament and of the Council, the content of mineral constituents must be considered as significant when 100 mL of a beverage can supply 7.5% of its reference value [22]. Except for Mo, this was fulfilled by all other elements (B, Ca, Cu, Fe, Mg, Mn, Ni, P and Zn) in all examined NDs.

Accordingly, consumption of NDs was established to contribute to RDAs of elements at maximum from 70.2% (for Mg, ND7) up to 291% (for Mn, ND7). Maximal fulfillments of TUULs were found to range from 7.6% (for Mo, ND7) to 87.4% (for Mn, ND7). The recommended intake of NDs (two bottles per day) could help in achieving more than 17% of RDAs for Mg, P and Fe, over 20% of RDA for Ca, and above 30% of RDAs for Cu, Zn, and Mn. According to outcomes achieved here, over 100% of minimal daily human needs, established for healthy middle-aged woman and man, could be completed by drinking examined nutritional dietary supplements in case of almost all elements indispensable for proper functioning of human organisms, i.e., Ca, Cu, Fe, Mn, P and Zn.

## 3. Materials and Methods

### 3.1. Samples and Reagents

Popular brand food nutritional products, widely distributed on the international and Polish market, were selected for this investigation. They were ready-to-drink, nutritionally complete, milk shake-style dietary supplements categorized as Food for Special Medical Purposes nutritional drinks (NDs). Four flavors (neutral, vanilla, chocolate and strawberry) of basic products and strawberry-flavored supplements enriched with proteins and fibers, as well as those for patients with chronic wounds and diabetes were chosen. Description of NDs examined in the present work is given in Table 5.

All experiments were carried out with the use of high purity deionized water (18.3 MΩ cm^−1^) obtained from an EASYpure™ system (Barnstead, Thermolyne Corporation, USA). All chemicals used in this study were at least of analytical grade and were verified for possible contamination.

All glassware and plastic bottles were washed with distilled water, cleaned with diluted nitric acid in an ultrasonic bath, and finally rinsed several times with deionized water.

### 3.2. Freeze-Drying

To concentrate analytes in analyzed NDs samples, they were freeze-dried. The whole content of a given original sample container (200 mL or 250 mL of nutritional drink) was placed in a wide-necked flask, weighted, sealed with a parafilm and placed in a refrigerator (−18 °C) for at least 24 h. Frozen NDs samples were lyophilized for 48 h using an ALPHA 1–2 LD plus freeze dryer (Martin Christ GmbH, Germany). Freeze-drying was performed at condenser temperature of −55 °C. Vacuum primary drying was carried out at –23 °C and 0.77 mbar. Containers with resulting dried products were weighted once again and weight losses were calculated. Finally, the dry material was removed from the flask, homogenized in an agate mortar, sealed in a polyethylene zip lock bag, and stored in a cool, dry and dark place until analysis.

### 3.3. Microwave-Assisted Mineralization

Accurately weighed portions of dried and powdered NDs samples (about 0.5 g) were transferred into Teflon digestion vessels. Next, 5.0 mL of concentrated nitric acid (65% HNO_3_, Suprapur, Merck KGaA, Darmstad, Germany) was added and kept overnight for pre-digestion. Afterwards, 1.0 mL of 30% H_2_O_2_ was added to each vial. Decomposition of pre-digested NDs samples was carried out in a microwave digestion system (Milestone, MLS-1200, MEGA), using a six-step program with a maximum power of 600 W. After finishing the digestion program and cooling the vessels, they were opened and clear and colorless digests were quantitatively transferred to 25.0 mL volumetric flasks and finally brought up to the volume with deionized water. Resulting sample solutions were stored at 4 °C prior to analysis.

Three parallel samples of each examined ND product were prepared and analyzed. Blank samples were simultaneously prepared through the complete sample preparation procedure for each set of digested NDs samples; they were analyzed and used for correction of results. Element analysis of blank samples confirmed high purity of chemicals used.

### 3.4. Measurements

Determinations of elements concentrations (Ag, Al, B, Ba, Ca, Cd, Co, Cr, Cu, Fe, Mg, Mn, Mo, Ni, P, Pb, Sr, Ti, V and Zn) were performed with a Jobin-Yvon ICP-OES instrument, model 38 S. Multielemental standard solutions were prepared for calibration, using an ICP multielement standard solution IV (Merck KGaA, Germany) and CertiPUR single standard solutions for Mo, P, Ti and V (Merck KGaA, Germany). The spectrometer was equipped with a parallel Burgener pneumatic nebulizer and a cyclone spray chamber for pneumatic nebulization sample introduction. During measurements, the following operating parameters were used for the instrument: the RF power of 1.0 kW, the plasma gas flow rate of 13 L min^−1^, the sheath gas flow rate of 0.2 L min^−1^, the nebulizer gas flow rate of 0.3 L min^−1^ and the sample flow rate of 1.0 mL min^−1^. The following analytical lines (wavelength in nm) were selected for quantification of analytes: Ag 338.3, Al 396.2, B 249.8, Ba 233.5, Ca 317.9, Cd 226.5, Co 228.6, Cr 267.7, Cu 324.8, Fe 259.9, Mg 285.2, Mn 259.4, Mo 202.0, Ni 221.6, P 214.9, Pb 220.4, Sr 407.8, Ti 334.9, V 292.4 and Zn 202.5.

Average total concentrations of determined elements in every ND sample were calculated based on results achieved for three parallel samples, separately preceded and analyzed, and expressed in mg per 100 mL of the analyzed ND.

Limits of Detections (LODs) was calculated for each element taking into account standard deviations of ten measurements performed for blank samples (SD_blank_) and slopes (S) of the calibration curves (according to equation: LOD = 3SD_blank_/S). Limits of quantification (LOQs) were calculated as multiplication of respective LODs (LOQ = 3.3LOD).

### 3.5. Method Validation

The NIST CRM of the non-fat milk powder (SRM 1549) was used to assess trueness of results obtained with the sample preparation procedure applied in this work, i.e., microwave-assisted mineralization, and as a repeatability control. Precision of measurements was calculated based on results of determinations done for individual samples of each material parallelly proceeded. Aqueous standard solutions, prepared by dilution of the ICP multielement standards (Merck KGaA, Germany), were used for calibration and to establish traceability of elements not included in the CRM. Five-point calibration curves within the concentration range from 0.100 to 5.00 µg∙g^−1^ were used for assessment of linearity. Respective correlation coefficients were within the range of 0.9931–1.000. LODs and LOQs for all measured elements were determined as described in paragraph 3.4.

### 3.6. Statistical Analyses

The two-tailed *t*-test at the 95% level of significance was applied to discover statistical significance of differences that occurred between total concentrations of elements determined in the CRM (NIST SRM 1549) and their certified values. The same test was used to verify differences between measured total concentrations of elements and data reported by the producer on nutritional products labels.

## 4. Conclusions

Concentrations of Ag, Al, B, Ba, Ca, Co, Cu, Fe, Mg, Mn, Mo, Ni, P, Sr, Ti and Zn in eight varieties of milk shake-style nutritional drinks for the special medical purpose were assessed. Generally, assessed contents of the elements were similar to values declared on product labels. For Mo in all analyzed NDs samples and several other elements in single samples, discrepancies between experimental results and stated contents were noted, most probably due to differences in composition of substrates used for formula fabrication. Our investigations indicated that nutritional drinks should be considered as a significant source of main and trace elements in the human diet, which along with the presence of trace metals such as Ag, Al, B, Ba, Ni, Sr and Ti indicates the requirement for strict control of mineral composition in nutritional drinks.

## Figures and Tables

**Table 1 molecules-25-00851-t001:** Results of element analysis of the non-fat milk powder standard reference material (NIST SRM 1549) ^a^.

Element	Concentration [µg/g]		*t*-Test ^b^
	Certified Value	Experimental Data	
Al	2 ^c^	2.13 ± 0.09	2.502, *N*
B	-	1.11 ± 0.11	
Ba	-	0.71 ± 0.01	
Ca	13000 ± 500	12400 ± 400	1.623, *N*
Cu	0.7 ± 0.1	0.75 ± 0.05	0.775, *N*
Fe	1.78 ± 0.10	1.80 ± 0.21	0.149, *N*
Mg	1200 ± 30	1090 ± 80	2.230, *N*
Mn	0.26 ± 0.06	0.24 ± 0.02	0.548, *N*
Mo	0.34 ^c^	0.33 ± 0.02	0.866, *N*
Sr	-	1.59 ± 0.14	
Zn	46.1 ± 2.2	45.3 ± 0.8	0.592, *N*

^a^ Mean value (*n* = 4) ± standard deviation (SD). ^b^ The calculated value of the *t*-test (t_calculated_) for α = 0.05, df = 3 + 3 – 2 = 4, and its critical value (t_critical_) equal to 2.776. ^c^ Information value. *N* The statistically insignificant difference.

**Table 2 molecules-25-00851-t002:** Concentrations of studied elements in nutritional drinks (NDs) in mg per 100 mL of the original sample.

Element	ND1	ND2	ND3	ND4	ND5	ND6	ND7	ND8
Ag	0.013 ± 0.007	0.015 ± 0.002	0.006 ± 0.001	0.010 ± 0.008	0.012 ± 0.002	0.010 ± 0.005	0.011 ± 0.009	0.003 ± 0.004
Al	0.141 ± 0.038	0.136 ± 0.014	0.097 ± 0.038	0.159 ± 0.016	0.169 ± 0.044	0.153 ± 0.035	0.065 ± 0.007	0.136 ± 0.039
B	0.800 ± 0.020	0.559 ± 0.028	0.246 ± 0.018	0.290 ± 0.021	0.342 ± 0.017	0.433 ± 0.037	<0.067 ^a^	<0.067 ^a^
Ba	0.013 ± 0.002	0.016 ± 0.002	0.017 ± 0.001	0.033 ± 0.008	0.020 ± 0.002	0.018 ± 0.002	0.012 ± 0.002	0.012 ± 0.001
Ca	159 ± 3	156 ± 2	153 ± 0	170 ± 4	332 ± 8	144 ± 1	290 ± 7	67.2 ± 1.2
Cd	<0.0003 ^a^	<0.0003 ^a^	<0.0003 ^a^	<0.0003 ^a^	<0.0003 ^a^	<0.0003 ^a^	<0.0003 ^a^	<0.0003 ^a^
Co	<0.0007 ^a^	<0.0007 ^a^	<0.0007 ^a^	<0.0007 ^a^	<0.0007 ^a^	<0.0007 ^a^	0.035 ± 0.008	0.034 ± 0.003
Cr	<0.0005 ^a^	<0.0005 ^a^	<0.0005 ^a^	<0.0005 ^a^	<0.0005 ^a^	<0.0005 ^a^	<0.0005 ^a^	<0.0005 ^a^
Cu	0.429 ± 0.036	0.391 ± 0.014	0.438 ± 0.037	0.403 ± 0.026	0.315 ± 0.010	0.375 ± 0.011	0.412 ± 0.004	0.113 ± 0.007
Fe	3.87 ± 0.18	3.69 ± 0.23	3.58 ± 0.01	3.34 ± 0.06	2.00 ± 0.18	3.48 ± 0.11	3.57 ± 0.17	1.68 ± 0.09
Mg	32.1 ± 1.3	34.7 ± 0.9	30.3 ± 0.0	35.7 ± 1.2	52.4 ± 2.2	29.2 ± 0.3	54.4 ± 0.2	26.3 ± 0.7
Mn	0.723 ± 0.020	0.732 ± 0.061	0.739 ± 0.066	0.677 ± 0.052	0.505 ± 0.018	0.694 ± 0.045	1.31 ± 0.04	0.430 ± 0.011
Mo	0.055 ± 0.003	0.052 ± 0.007	0.041 ± 0.007	0.047 ± 0.003	0.044 ± 0.025	0.037 ± 0.009	0.038 ± 0.005	0.032 ± 0.002
Ni	0.063 ± 0.012	0.031 ± 0.011	0.028 ± 0.003	0.044 ± 0.020	0.036 ± 0.002	0.054 ± 0.005	<0.003 ^a^	<0.003 ^a^
P	174 ± 4	160 ± 5	182 ± 5	172 ± 9	269 ± 15	156 ± 9	214 ± 11	56.1 ± 1.3
Pb	<0.008 ^a^	<0.008 ^a^	<0.008 ^a^	<0.008 ^a^	<0.008 ^a^	<0.008 ^a^	<0.008 ^a^	<0.008 ^a^
Sr	0.062 ± 0.007	0.070 ± 0.003	0.085 ± 0.009	0.079 ± 0.005	0.135 ± 0.015	0.067 ± 0.007	0.076 ± 0.003	0.042 ± 0.005
Ti	0.014 ± 0.004	0.007 ± 0.001	0.006 ± 0.001	0.006 ± 0.002	0.009 ± 0.010	0.006 ± 0.002	0.005 ± 0.002	0.004 ± 0.002
V	<0.0008 ^a^	<0.0008 ^a^	<0.0008 ^a^	<0.0008 ^a^	<0.0008 ^a^	<0.0008 ^a^	<0.0008 ^a^	<0.0008 ^a^
Zn	2.56 ± 0.03	2.73 ± 0.28	2.43 ± 0.09	2.38 ± 0.21	2.33 ± 0.09	1.99 ± 0.14	4.85 ± 0.09	1.21 ± 0.03

^a^ Below the limit of quantification (LOQ).

**Table 3 molecules-25-00851-t003:** Declared and determined concentrations of Ca, Cu, Cr, Fe, Mg, Mn, Mo, P and Zn (in mg L^−1^) in nutritional drinks (NDs).

	Experimental				Declared	Experimental	Declared	Experimental	Declared	Experimental	Declared	Experimental	Declared
	ND1	ND2	ND3	ND4	ND1-ND4	ND5	ND5	ND6	ND6	ND7	ND7	ND8	ND8
Ca	159	156	153	170	*174*	332	*350*	144	*174*	290	*225*	67.2	*53*
Cu	0.429	0.391	0.438	0.403	*0.43*	0.315	*0.35*	0.375	*0.43*	0.412	*0.68*	0.113	*0.18*
Cr	<0.0005	<0.0005	<0.0005	<0.0005	*0.014*	<0.0005	*0.013*	<0.0005	*0.014*	<0.0005	*0.012*	<0.0005	*0.012*
Fe	3.87	3.69	3.58	3.34	*3.8*	2.00	*2.1*	3.48	*3.8*	3.57	*3.0*	1.68	*1.6*
Mg	32.1	34.7	30.3	35.7	*33*	52.4	*55*	29.2	*33*	54.4	*42*	26.3	*23*
Mn	0.723	0.732	0.739	0.677	*0.80*	0.505	*0.63*	0.694	*0.80*	1.31	*1.30*	0.430	*0.33*
Mo	0.055	0.052	0.041	0.047	*0.024*	0.044	*0.020*	0.037	*0.024*	0.038	*0.019*	0.032	*0.010*
P	174	160	182	172	*174*	269	*300*	156	*174*	214	*182*	56.1	*47*
Zn	2.56	2.73	2.43	2.38	*2.9*	2.33	*2.4*	1.99	*2.9*	4.85	*4.5*	1.21	*1.2*

**Table 4 molecules-25-00851-t004:** The assessed nutritive value of nutritional drinks (NDs) compared to recommended daily dose values.

	RDA, mg/day [18,19]	TUUL, mg/day [20,21]	Nutritional Norms Fulfillment Related to Consumption of 2 Bottles of NDs, %	
			Calculated for RDA	Calculated for TUUL
B	-	11–20	-	3.08 (ND3)–18.2 (ND1)
Ca	1000–1300	2500	20.7 (ND8)–116 (ND7)	10.8 (ND8)–46.3 (ND7)
Cu	0.9–1.3	8–10	34.8 (ND8)–183 (ND7)	4.52 (ND8)–20.6 (ND7)
Fe	10–27	40–45	18.5 (ND5)–143 (ND7)	11.1 (ND5)–35.7 (ND7)
Mg	310–420	350	17.4 (ND6)–70.2 (ND7)	20.9 (ND6)–62.2 (ND7)
Mn	1.8–2.6	6–11	48.6 (ND5) – 291 (ND7)	11.5 (ND5)–87.4 (ND7)
Mo	0.045–0.050	2	0.19 (ND6)–0.34 (ND7)	4.63 (ND6)–7.64 (ND7)
Ni	-	0.6–1.0	-	0–26.2 (ND1)
P	700–1250	3000–4000	18.0 (ND8)–122 (ND7)	5.61 (ND8)–28.5 (ND7)
Zn	8–13	23–40	37.2 (ND8)–242 (ND7)	12.1 (ND8)–84.3 (ND7)

RDA Recommended dietary allowance. TUUL Tolerable uptake upper level.

**Table 5 molecules-25-00851-t005:** Examined nutritional drinks (NDs).

Sample	Flavor	Energy, kcal/100 mL	Proteins Contents, g/100 m	Fibers Contents, g/100 mL	Intends
ND1	Neutral	240	9.6	-	A, B, C, D, E, F, G, H
ND2	Vanilla	240	9.6	-	A, B, C, D, E, F, G, H
ND3	Chocolate	240	9.6	-	A, B, C, D, E, F, G, H
ND4	Strawberry	240	9.6	-	A, B, C, D, E, F, G, H
ND5	Strawberry	240	14.4	-	A, B, C, D, H
ND6	Strawberry	240	9.6	3.6	A, B, C, D, E, F, H
ND7 ^a^	Strawberry	124	9.0	-	I
ND8	Strawberry	102	4.9	2.0	J

A, Short bowl syndrome. B, Intractable malabsorption. C, Preoperative preparation of undernourished patients. D, Inflammatory bowel disease. E, Total gastrectomy. F, Dysphagia. G, Bowel fistulae. H, Disease related malnutrition. I, Chronic wounds and bedsores. J, Diabetes and impaired glucose tolerance. ^a^ Enriched with arginine (at 1.5 g/100 mL), zinc and antioxidants.

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
