# Peer review of "Mineral Constituents Profiling of Ready-To-Drink Nutritional Supplements by Inductively Coupled Plasma Optical Emission Spectrometry"

_molecules, 2020, doi:10.3390/molecules25040851_

Round 1
Reviewer 1 Report
Authors' manuscript (MS) is based on valuable analytical work, but the presentation of the results should have been somewhat better. The style is quite 'heavy', which means that the reading might be difficult, and the main message of MS is obscure. My main suggestion would be – try to shorten MS. Some 30% of it is redundant.
Abstract: the first sentence should be modified, try to divide it into 2 shorter sentences, pay attention to the style. If it sounds boring or confusing, it will immediately repel readers. And that is the problem with this MS – it will repel them as you stuffed too much story into this MS which is not necessary. It would be recommended if you give MS to native speakers or language professionals to edit it.
Line 17: what does dens mean? Highly concentrated?
It is obvious – I ask, what is obvious? Based on what? You have to reformulate the sentence (pay attention to scientific style of writing which is quite uniform, look at other articles published in high IF journals for examples). Daily human needs – do you refer on RDI (ref. daily intake)? Nutridrinks – please, use only one term for your samples, not 2-3 of them, and use it uniformly throughout the text. 30% - please, provide some statistical parameter, not just number. Not main but major (elements). Lines 20-24 are quite confusing (especially the last one!). This part must be improved. Interrelationships – I am not happy with this term. I would suggest you to use correlations. Did you check normality of data distributions? Pearson coefficient is used for normally distributed data (what was the number of data per element?).
Lines 29-48: please, carefully inspect this part which is very important (lots of readers will give up if this part is heavy for reading). The first few sentences should be devoted to the main object of your research – nutritional supplements. Give some intriguing facts so that you capture readers' attention.
Table 2 should be edited (decrease font as few numbers overlap), units are given in the text, but they must be shown in table title; what was the number of measurements (n) per element?
Line 151-152: please, reformulate this (smallest number of elements were determined): you have to provide statistical evidence (LOQ, LOD, etc., based on them you can say which elements were not able to determine due to their levels below LOD, etc.). 158-159: your data are expressed in certain units, while you refer on literature data expressed in different units – please, recalculate them, so that it is possible to compare them. 161: you claim that the highest levels of ALL (!) examined elements were found for the sample ND1. Are you sure? I can not agree with you. Please, clarify that.
184: European market; why didn't you start your discussion with comparison of your data in that regard? Briefly, you measured selected elements in several drinks. First, we would all be curious about compliance of those drinks with EU regulative. Then compliance with stated levels on labels. Then comparison with other similar (relevant) publications. 190 – alarming – please, use different term (data surpassing regulative levels?), this one is not recommended; was something alarming in your data or..?
Could you provide figure instead of table 2? Please, do not repeat data which can be found in table. You have to put them in context, interpreting them. Please, put more efforts into MS, so that it sounds interesting to international readership.
Reviewer 2 Report
The work concerns the mineral profiling of selected commercial milk shake style of nutritional drinks as the important part of quality control and monitoring investigated food supplements. The manuscript is well written however some authors should consider same comments before publishing in the molecules journal.
Detailed comments;
point 2. 1, 88: is accuracy should be “truness”. Accuracy = truness and precision; table 2 should be below the detection limit. Important!!! Specify LOD value.LOD is determined by various methods, while LOQ is the most cammon mathematical conversion LOD. The authors should indicate what method they determined LOD.
How established traceability for the other elements that were not certified in the CRM used? For elements that cause toxic effect, for example Cd, Pb, Ni, Cr , are particularly important.
Material and Methods.Validation parameters are missing. Point : “Figure of Merit”.
After completing the remarks, the work may be published in the journal Molecules.
Round 2
Reviewer 1 Report
Abstract: what do numbers (mean -5.91, and sd 14) mean? units are missing. I am confused with this.
Table 2: what does 'a' signify?
Instead of the Pearson coefficient, why not using a nonparametric correl. coefficient (Kendal tau, or Spearman, etc.)?